# Role of Drone Technology Helping in Alleviating the COVID-19 Pandemic

**DOI:** 10.3390/mi13101593

**Published:** 2022-09-25

**Authors:** Syed Agha Hassnain Mohsan, Qurat ul Ain Zahra, Muhammad Asghar Khan, Mohammed H. Alsharif, Ismail A. Elhaty, Abu Jahid

**Affiliations:** 1Optical Communications Laboratory, Ocean College, Zhejiang University, Zheda Road 1, Zhoushan 316021, China; 2Department of Biomedical Engineering, Biomedical Imaging Centre, University of Science and Technology of China, Hefei 230009, China; 3Department of Electrical Engineering, Hamdard Institute of Engineering & Technology, Islamabad 44000, Pakistan; 4Department of Electrical Engineering, College of Electronics and Information Engineering, Sejong University, Seoul 05006, Korea; 5Department of Nutrition and Dietetics, Faculty of Health Sciences, Istanbul Gelisim University, Istanbul P.O. Box 34310, Turkey; 6School of Electrical Engineering and Computer Science, University of Ottawa, 25 Templeton St., Ottawa, ON K1N 6N5, Canada

**Keywords:** UAVs, COVID-19, technologies, healthcare, applications

## Abstract

The COVID-19 pandemic, caused by a new coronavirus, has affected economic and social standards as governments and healthcare regulatory agencies throughout the world expressed worry and explored harsh preventative measures to counteract the disease’s spread and intensity. Several academics and experts are primarily concerned with halting the continuous spread of the unique virus. Social separation, the closing of borders, the avoidance of big gatherings, contactless transit, and quarantine are important methods. Multiple nations employ autonomous, digital, wireless, and other promising technologies to tackle this coronary pneumonia. This research examines a number of potential technologies, including unmanned aerial vehicles (UAVs), artificial intelligence (AI), blockchain, deep learning (DL), the Internet of Things (IoT), edge computing, and virtual reality (VR), in an effort to mitigate the danger of COVID-19. Due to their ability to transport food and medical supplies to a specific location, UAVs are currently being utilized as an innovative method to combat this illness. This research intends to examine the possibilities of UAVs in the context of the COVID-19 pandemic from several angles. UAVs offer intriguing options for delivering medical supplies, spraying disinfectants, broadcasting communications, conducting surveillance, inspecting, and screening patients for infection. This article examines the use of drones in healthcare as well as the advantages and disadvantages of strict adoption. Finally, challenges, opportunities, and future work are discussed to assist in adopting drone technology to tackle COVID-19-like diseases.

## 1. Introduction

At present, the new coronavirus disease 2019 (COVID-19) has unpredictably spread across the globe and severely impacts human lives, research activities, economies, and industries across the globe. Because of its severe impacts and infectious behavior, management and healthcare have become crucial issues despite innovative treatment, vaccines, and medical facilities [1]. Several research initiatives have been carried out to handle this disease through emerging technologies including the Internet of Medical Things (IoMT), robots [2], artificial intelligence (AI) [3], and unmanned aerial vehicles (UAVs) [4]. Disruptive wireless, automated and digital technologies such as AI, machine learning (ML), deep learning (DL), and deep neural networks (DNNs) have shown potential in combating the coronavirus [5,6]. Automated technologies such as UAVs and robots with the potential to carry heavy payloads have been greatly used to smartly control the spread of this virus by minimizing human contact in different environments. These technologies can assist in medical testing, treatment, vaccine delivery, and accurate diagnosis.

Several robotic technologies have been deployed to assist medical staff in virus symptom detection, assist infected patients, and minimize the further spread of the virus [7]. These promising technologies are helpful in social and physical distancing and virus containment. These technologies can be used in any challenging environment where human activity is dangerous [8], as this virus has made the public realm into a crucially hostile environment. Potential intersections between COVID-19 management and robotic technologies are based on the following [8]:

Minimizing human-to-human contact: Robots are used to reduce human activity along with ensuring cost-effectiveness, reliability, and efficiency in performing different tasks such as social services, military operations, healthcare, and logistics. Robots can minimize human functions to control pandemics by reducing virus transmission, managing social distancing, controlling human contact, and operating autonomously.

Managing, monitoring, and controlling mobility: Quarantine, lockdown, and controlling human mobility play a major role in preventing the further spread of this disease. AI, ML, and robotic technologies can significantly monitor human mobility and avoid human gathering by territorial control at different checkpoints.

Similarly, UAV technology has proven its stature to curb the risk of COVID-19 [9]. Several countries including China, USA, Japan, and Australia have been using this technology to combat this virus. For instance, China has been using more than 100 UAVs to perform surveillance tasks in various cities [10]. This approach has proved to be useful to avoid virus transmission and ensure social distancing. Several other countries are considering UAVs for vigilance, governance, delivering food and medical supplies, thermal scanning, monitoring, and sanitization. In the US, UAVs have been deployed to deliver medical supplies such as personal medical kits for COVID-19 to distant places [10]. In Australia, UAVs have been used to monitor patients with doubtful virus symptoms [11]. Sensors integrated into UAVs are used to monitor respiratory rate, heartbeat, body temperature, and other abnormalities. In addition, effective results in combating this virus can be achieved by integrating UAVs with other technologies such as thermal imaging, AI, ML, DL, and IoT.

### 1.1. Scope and Contributions

The key aim of this review article is to assist is to educate readers who are interested in deploying UAV technology to combat COVID-19. Keeping this interest in mind, this article presents a brief overview of UAV technology and its potential to curb the risk of COVID-19 and reviews the state-of-the-art UAV technology in numerous applications, including spraying disinfection, transmitting messages through QR codes or loudspeakers, delivering medical supplies, surveillance, inspection, screening, and detection. This paper also examines practical approaches for combating COVID-19 through the integration of promising technologies including AI, ML, DL, IoT, and edge computing. Extensive assessments are conducted for several feasible solutions leveraging technologies including robotics, sensors, wearable devices, and virtual reality (VR). This study concludes with an analysis of the present problems, security issues, potential, and recommendations for future usage of these technologies for COVID-19 and similar diseases.

### 1.2. Organization of the Paper

This paper is organized as follows: Section 2 is focused on the related research contributions to UAVs to combat COVID-19. Section 3 provides basic information about UAVs. Section 4 addresses different applications of UAVs to handle COVID-19. Section 5 is devoted to highlighting the role of different emerging technologies in minimizing the risk of COVID-19. Several open challenges, security issues, opportunities, and future recommendations are briefly explained in Section 6. Finally, Section 7 concludes the paper.

## 2. Related Work

A human viral illness called coronavirus disease 2019 (COVID-19) is associated with significant respiratory distress. Because of the effects of COVID-19 infection and its outbreak on human health, the World Health Organization (WHO) has designated it as a Public Health Emergency of International Concern (PHEIC) as of January 30, 2020 [12]. Given that infection spreads quickly across large populations, COVID-19 has been classified as a pandemic illness. The greatest way to prevent its transmission, according to WHO, is to take precautions as soon as possible. Healthcare organizations are primarily concentrating on implementing extensive testing and successfully treating sick people in order to combat the pandemic. Additionally, countries agree that enforcing lockdowns in infection-prone areas is the greatest strategy for reducing infection and bringing sustainability to the present pandemic crisis. Therefore, a specific framework for reshaping society is needed in order to execute the lockdown while still fulfilling the requirements of the healthcare system. In these conditions, smart technologies have had an influence on a number of areas, including healthcare, government, and business. The use of technologies such as telemedicine [13], artificial intelligence (AI) [14], robots [7,8], and drones [4] has increased significantly throughout the pandemic to bear the burden of frontline health system soldiers. Robotic and drone technology has been set up to help with monitoring, early infection screening, transportation of medical supplies, disinfection, and other processes. Robotics has also been extensively used to provide patients and healthcare professionals with medical advice. AI has been used in forecasting areas with infection concerns, prescribing medication, computing diagnosis findings, and other applications because of its capacity to operate independently, self-learn to improve with the addition of fresh data, and draw imperatives in short periods of time. Even the mobile phone network is seen as a crucial tool for combating the pandemic condition. Mobile phone technology is now widely used in cities and digitally connects people to bigger populations. Smartphone apps (Apps) have been used as the ideal model to interact with wider populations since mobile phone networks have been compared to the veins of cities. Numerous smartphone apps have been created to raise awareness, offer medical help, and most importantly, stop the COVID-19 virus from spreading by tracing contacts. Authorities are concentrating on the adoption of protective measures and quick testing techniques as we prepare to walk out and restore normalcy to everyday life. In this sense, the usage of face masks has shown to be successful in reducing the COVID-19 virus’s ability to propagate. Due to the unexpected increase in mask demand, new advancements in mask manufacturing technology have been made. A defensive line against COVID-19 has been established thanks to the collaboration of several industries, the use of technology, and competent governance. Understanding how these advancements are working together to tackle the pandemic crisis is crucial given the technology’s astounding contribution to the fight against COVID-19 infection. In Table 1, pandemic- and disaster-related UAV-based systems are summarized.

### The Crisis due to the COVID-19 Pandemic

The marketing of many consumables has been disrupted, and several locations have seen the closure of stores selling non-emergency supplies. Large firms are being requested to provide masks, ventilators, kits, and other essential forms of equipment to assist in avoiding this sickness since there is a scarcity of medical supplies. Nevertheless, they are dealing with a major logistical problem [24].

Among the worst-affected industries is the tourism and travel sector. Here, the majority of domestic and international travel has been halted, and travelers have postponed their plans for important events throughout the globe. Many businesses are closing, and in certain countries, television shows are even being produced. There have been several postponements of international athletic events [25]. The governance frameworks are under scrutiny. People are frustrated and outraged with the situation, especially with those who are in charge of it. They dispute the significant spending on arms and research support for the armament business, arguing that if these funds had been allocated to public health, a pandemic of this nature would not have happened. This rage can occasionally be directed against those who are actually responsible for the transmission of this disease. Additionally, we observe that catastrophe managers are the targets of people’s rage. People are prepared to do nothing and wish to avoid infection. The general public must stay away from crowded areas and public transportation. Sadly, social media spreads false information and misinformation, which demotivates individuals. Additionally, social events such as marriages, vacations, weddings, and other leisure activities are postponed or canceled [26,27]. Transportation and logistics are a major problem during this epidemic.

## 3. Unmanned Aerial Vehicles (UAVs)

Unmanned aerial vehicles (UAVs), also referred to as drones, have received momentous attention in various domains of civil and military operations because of their high mobility, enhanced stability, low cost, and high endurance in multiple tasks. The use cases of UAVs are extensively growing due to the integration of various emerging technologies such as 5G/B5G, artificial intelligence, Internet of Things (IoT), and mobile edge computing. UAVs are being used in a wide variety of applications such as logistics, forest monitoring, construction, freight transportation, communication, healthcare, post-disaster operation, search and rescue, remote sensing, agriculture precision, power-line inspection, traffic surveillance, and object detection and tracking [28,29,30].

In general, UAVs are controlled aerial vehicles which do not carry any human pilot. UAVs can be operated autonomously and remotely through microprocessors, sensors, and other equipment [31]. UAVs make use of communication links to connect with satellites or ground control stations (GCSs) such as smart phones or laptops. To perform remote operation, a human operator is required to control the UAV using a remote control. The use of UAVs is highly encouraged in scenarios where human intervention is strictly limited or hazardous.

Due to the overwhelming interest in drones [32,33,34], several UAVs of varying sizes and forms (shown in Figure 1) have been developed to perform a variety of tasks. The categories of UAVs—single-rotor, multi-rotor, fixed-wing, and hybrid UAVs—have been used in the pandemic and other industrial applications. Each UAV has advantages and disadvantages that help us choose the one that is best suited for the application as shown in Table 2. Different characteristics of these UAVs are also summarized in Table 3.

## 4. Applications of UAVs during COVID-19

UAVs have shown great potential in smart cities worldwide. Smart cities have smart healthcare systems based on telemetry, implantable medical equipment, and medical drones to quickly deliver first-aid supplies. UAVs have proven their stature to tackle the COVID-19 pandemic in different countries. However, it is worth noting that the leading organization tackling COVID-19 is the national EMS institution along with several parties such as EMS personnel, nurses, and medical doctors. In addition, several policymakers are considering different preventive measures to fight against COVID-19 including wearing surgical masks, avoiding facial touching, regular handwashing, city lockdown, high-risk area avoidance, social gathering avoidance, and implementing health codes. Policymakers should consider the balance between the economy and public safety before introducing any new measures. Currently, UAVs are being used to perform various tasks (as shown in Figure 2) to prevent COVID-19 such as:Transport of vaccines and medical kits;Public announcements;Crowd surveillance;Spraying disinfection;Mass screening;Crown aerial monitoring;Delivering vaccines and other medical supplies.

**Figure 2 micromachines-13-01593-f002:**
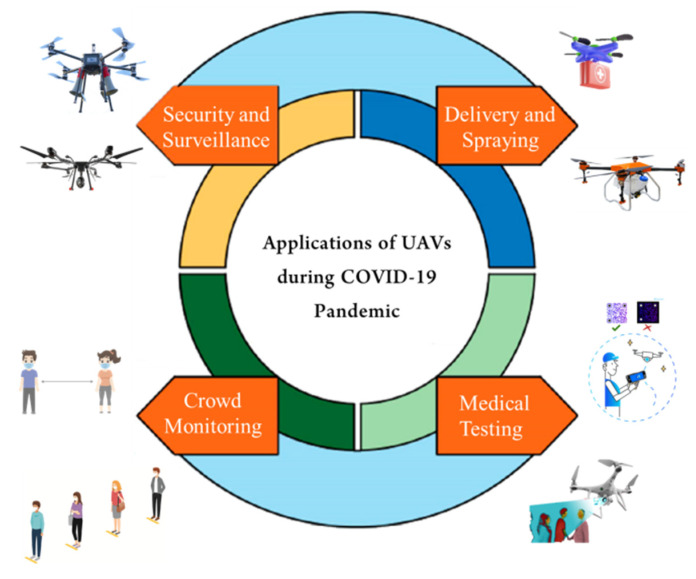
Role of UAVs in COVID-19 pandemic.

### 4.1. Surveillance and Inspection

The most common usage of drones is for area monitoring or aerial surveillance, both in routine operations and during special operations to battle COVID-19. Area monitoring or aerial surveillance was employed in every country where drone applications were recorded. When visually watching a certain region, several factors must be taken into consideration, including the area’s size, topographical features, spaciousness, built-up ratio, locations of particular importance within the area, the approach points of the area, and the duration of the observation. Point monitoring may be adequate in situations involving smaller or transparent open spaces; in situations involving larger areas, the deployment of many drones arranged in a network is needed; and in situations involving part-time observation, it is essential to patrol with one drone [4]. Drones could even keep an eye on things where there is not much human involvement. The effort and time needed to keep an eye on these places are too great. The invasion of private rights is one of the main obstacles to the drone-based monitoring of individuals. However, by releasing recommendations on public interests, governments are attempting to uphold these rights.

Since 2019, COVID-19 has been mysterious; it has impacted a number of health professionals, including doctors, nurses, cleaning staff, and security personnel. Security staff may monitor unwanted movements and follow activity in hotspot regions without physically visiting them by using drones. They have the ability to take pictures of alleged violators of the occasional preventive instructions provided by governments. There are several benefits to performing this type of surveillance using drones, including avoiding COVID-19 infection among security personnel; saving gasoline and other resources that are often used for physical surveillance; and anytime, anyplace surveillance. Police officers are reluctant to visit hotspots in the absence of drones, but this is a necessary aspect of their job and they must. Since it is the responsibility of the government to protect security personnel from COVID-19, most of them are giving their monitoring departments’ access to drones. In order to include multiple flying ad hoc network (FANET) devices in the COVID-19 task teams, government authorities are modifying their rules. Together with monitoring human mobility [36], security personnel can provide people advice on how to avoid crowding, minimize one-on-one interaction, emphasize the value of social distance, etc. To perform these tasks, the authors of [37] used a drone swarm in a specific area. A camera and sensor were mounted on the ground to capture videos and a real-time view of that area. The medical staff operating through GCS can trace infected people through thermal cameras. Cameras can also monitor social distancing, as shown in Figure 3. Drones can gather information from the GCS and can deliver it to the public through voice or display screens. Table 4 summarizes drone technology used for surveillance tasks in different countries.

### 4.2. Broadcasting Messages Using Loudspeaker on Board

In order to spread information, drones equipped with speakers, flags, QR codes, etc., are essential, especially during epidemics. Authorities can use these drones to some extent to deliver crucial messages, particularly in areas where communication links are down. These drones can prove to be one of the most effective communication tools during lockdowns. Municipal corporations can use these drones with sky speakers to reach out to residents in their areas with important messages (such as updates about a lockdown extension, information about which stores are allowed and which are not, health-related communications, implementation of social distancing, and other precautionary guidelines). Due to the fact that most people rely on TV for their communications, especially in rural regions, this sort of communication can break down barriers between municipal corporations and the populace. Figure 4 shows how a UAV with a speaker works to provide alerts to the public and keep individuals safe at home. Drones with speakers are now being used by numerous nations, especially the US, Spain, Britain, and China to communicate with those affected by the COVID-19 epidemic.

The usage of a loudspeaker must consider loudspeaker efficiency and interference, the quality of voice transmission, the pitch and speech segmentation employed, and the space between the flying location of the drone and the individual or group to be briefed. However, entering a public zone belonging to a person is already evidently an issue of flight safety; keeping a minimum safe spacing of 5 m is justifiable. At longer drone-to-human distances, it may be considered that the target’s information receipt is instructive; at relatively short distances, it is a warning, and in the proximity of the open zone, it is already threatening. The action of a drone that maneuvers unexpectedly or dramatically in close proximity and communicates instantly, potentially at an excessive level, may be meant as a warning but may be perceived as menacing.

These speaker-capable drones can aid with crucial communications, alerts during hazardous tasks, or rescue missions in addition to just chatting. Such drones can show themselves to be effective weapons in vast, crowded, and challenging locations while dealing with calamities such as floods, earthquakes, and fires. Despite these benefits, during emergencies and epidemics, they could keep authorities away from close contacts, preventing any likelihood of infection from risky actions and ensuring that the authorities can confidently perform their crucial jobs. Table 5 summarizes drone technology used to broadcast messages in different countries to implement COVID-19 precautionary measures.

**Table 5 micromachines-13-01593-t005:** Drone technology used to broadcast messages for COVID-19.

Reference	Country	Purpose
[43]	USA	To broadcast a warning to suspected people to follow social distancing rules
[44]	Spain	Police used drones to yell at the public for being outside and ignoring the lockdown
[45]	China	Drone mounted with a speaker to inform the public to wear a mask
[46]	Malaysia	To give announcements and alerts to the public to curb COVID-19 spread
[47]	Hungary	To inform the public to stay at home due to COVID-19
[48]	USA	Anti-COVID-19 volunteer drone to ensure social distancing of 6 feet
[49]	Rwanda	To broadcast a crucial message about COVID-19

**Figure 4 micromachines-13-01593-f004:**
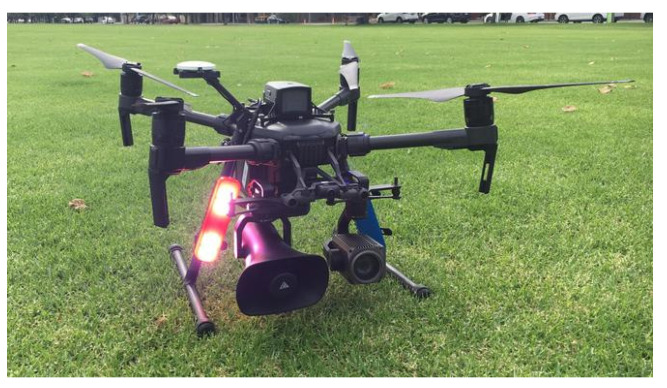
Broadcasting a message by using a loudspeaker [50].

### 4.3. Broadcasting Messages by Towing QR Code Flag and Illumination

There are various examples of using a drone tow a QR code banner in place of a loudspeaker to facilitate communication. You can use a QR code flag in addition to or rather than a loudspeaker. While QR codes appear to be a “static” version of communication, loudspeakers appear to be a “dynamic” version since they may broadcast “live” messages to the whole public. In the first scenario, the targeted population is being addressed by the drone operator or mission commander who can respond to the situation with brief messages right away in a dynamic environment. In the second scenario, the environment is more or less static and the operator or commander must deliver more complex but standard messages. In the latter scenario, it goes without saying that the targeted demographic will need to utilize their cell phones to obtain the required information online. In many locations, such as crossings, mall parking lots, congested roads, and large-scale sporting events where the border or gate crossing procedure or the staying situation might vary daily, using a QR code can be useful.

On the other side, Chinese authorities are utilizing UAVs to scan drivers entering Shenzhen city by placing overhead QR codes [51]. Figure 5 depicts a motorist who is returning to Shenzhen registering by scanning the QR code. These UAVs were flown close to highway exits or toll booths to encourage approaching vehicles to register so that authorities could follow their movements around the city. However, the drone must be in close proximity, and this application can also cause challenges such as delays and traffic congestion. Additionally, it must be smartly controlled and it should be very close in order to avoid any physical or economic hazards.

In 2020, 300 drones were used to illuminate the sky to honor frontline heroes, medical workers, and patients in the Netherlands [52]. In Seoul, South Korea, hundreds of drones performed a show displaying messages to support the country in the fight against COVID-19. The show started with showing precautionary steps and ended with showing gratitude to medical personnel and the public for their collective efforts against COVID-19, as shown in Figure 6.

### 4.4. Disinfecting Surfaces and Common Areas

As COVID-19 instances increase at an enormous speed, it is clear that most cases are brought on by surface contact with objects found in public spaces, such as chairs, tables, railings, door handles, elevator buttons, public transportation, and shopping centers. Depending on the environmental conditions, the COVID-19 virus can remain present on various surfaces for a few hours to many days. The first step in preventing secondary transmission is to sterilize these surfaces and locations. In general, air disinfection and surface disinfection are the two most popular methods to improve disinfection. In comparison to air decontamination, surface decontamination was more successful in managing the COVID-19 epidemic.

Air disinfections, on the other hand, can offer psychological relief and emotional reassurance that the authorities are making an effort to handle the crisis, and they can also keep rats away. Among the several methods for air sanitization, drone spraying has proven to be beneficial. According to their power and tank capacity, spraying drones often come in several variants. A generalized UAV spraying concept with a small tank is shown in Figure 7. While sanitization employees may be exposed to viral diseases while performing their vital jobs, these UAVs are simple to operate and mobilize and prevent the operator from becoming ill. On the other hand, although having a number of benefits, air disinfection has not been shown to be a reliable way to stop the transmission of viruses. Alcohol-based and other chemicals should not be sprayed excessively on people, cars, or the environment since they are extremely dangerous. Drones can be used to spray disinfectant into public areas when no one is there, such as at the end of the day or on vacations. Sprays based on agriculture should never be used since they are more dangerous than air disinfectants.

Recently, several studies have been reported on employing drones to spray disinfecting solutions against novel coronaviruses [54]. It is important to note further scientific studies that are creating strategies to handle the COVID-19 epidemic more expertly. The study by Kumar et al. [9] focused on interconnected drone-based systems, such as the potential use of drones for disinfectant spraying. In this trial, a 2 km radius was sprayed in 10 min; however, there was no information about the circumstances. Alsamhi et al. [55] provided a framework and suggested a mechanism for employing several drones in decentralized ways. The use of drones to spray disinfectant was also addressed in this study, although no specifics about the practice were provided. In [56], the authors used UAVs to perform disinfection tasks in some areas. They focused on the evaluation of static and dynamic behavior, as well as the impact of spraying flow, mission speed, and flying height. The recommended feasible height was 3 m for good performance. Table 6 summarizes drone technology used to spray disinfectants in some countries against COVID-19.

**Table 6 micromachines-13-01593-t006:** Drone technology used to spray disinfectants for COVID-19.

Reference	Country	Purpose
[57]	China	To spray disinfectants to stop the spread of COVID-19
[58]	India	To improve the efficiency and speed of the sterilization operations in public places
[59]	Spain	Spain’s military used agriculture drones to spray disinfectants to stop the spread of COVID-19
[60]	USA	Drone to spray disinfectants in large places
[61]	UAE	Dubai Municipality conducts a massive sterilization drive against COVID-19 through drones

**Figure 7 micromachines-13-01593-f007:**
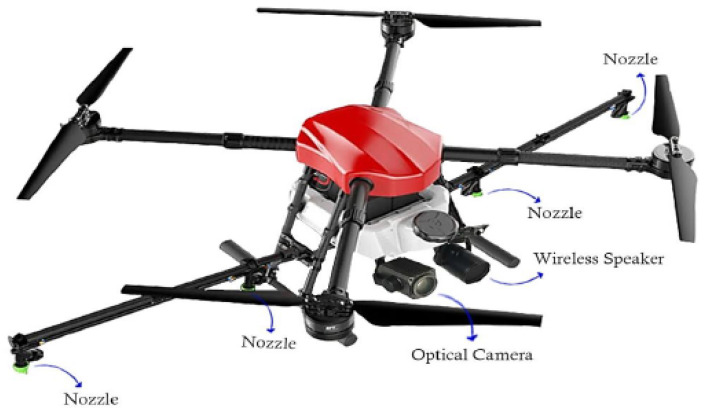
Drone used for spraying disinfectants [62].

### 4.5. Supporting Medical Services

By using drones, delivery services for food, parcels, medical supplies [63], etc., may become quicker and more sustainable at a lower cost than more conventional delivery methods such as vans and vehicles [64]. Manual delivery may be replaced by the delivery method’s transformation. In order to increase delivery effectiveness and quality of service, using drone control may make it possible to accomplish automatic, unmanned, and information-based distribution. This will close the gap that exists between orders and the capacity for delivery services.

Healthcare professionals would be one of the prominent stakeholders in managing and providing the services to the general public and are at the forefront of modern acceptance of technology. Healthcare professionals have a favorable view regarding the use of drones, according to a study conducted in Oslo, Norway [65]. It is more crucial than ever to transport vaccines to remote regions in a quick, secure way, especially in the COVID-19 age. Drones may be used to deliver COVID-19 vaccines to inaccessible locations, according to several research studies devoted to this topic [66,67]. During the epidemic, a few studies examined customer views concerning drone delivery of general goods or food; these studies, therefore, did not examine public approval of drone delivery in general [68]. Therefore, it is important to investigate how healthcare professionals feel about using drones to deliver medications and vaccines during the COVID-19 pandemic. Malaysia is actively investigating the use of drones to deliver medications and vaccines. Nevertheless, studies on the adoption of drones as a method of delivering vaccines and medications to rural regions by healthcare providers are scarce. In a recent study [69], the authors evaluated the feasibility of drones to deliver medical supplies considering multiple aspects such as payload integration, fleet development, and medical delivery simulations. In [70], the authors used drones for drug distribution considering an optimum way. They focused on finding an optimized trajectory for drones to deliver a drug. They proposed a hybrid model containing K-mean clustering and ant colony algorithms. They compared the performance of different algorithms and stated that the ant colony algorithm yields the closest to optimum results. Drones used to deliver medical and food supplies during COVID-19 are shown in Figure 8. Table 7 summarizes the drone technology for delivery services in different countries.

Massachusetts Institute of Technology (MIT) and UNICEF are working in collaboration to determine the impact of the use of drones to deliver vaccines in remote communities of Nepal [71]. They also analyzed the cost and advantages of drone usage in two districts and developed a set of recommendations. To ensure the complete availability of vaccines, they identified the optimal drone system, delivery hubs, payload, and range, showing a transportation cost saving of USD 0.10–0.12 compared to baseline. The analysis reveals that cost savings can be achieved only when capital costs, including infrastructure cost, delivery start-up, and equipment procurement, are subsidized. This economic analysis shows an intriguing investment case and introduces a sustainable path ahead to integrate drone technology into health supply chains. A Deloitte report and the 2021 World Economic Forum revealed that drone technology can be a cost-efficient option for governments if they have: large-scale deployment, expensive ground transportation, and an affordable drone vendor [72]. Thus, the implementation of drone technology can decrease cost and enhance vaccine availability in a wide range of circumstances and settings if this technology is extensively used to overcome the capital costs of system deployment and maintenance. The major drivers to ensure cost-effectiveness are distance to be traveled, people to be vaccinated, and road speed and cost of conventional ground vehicles [73]. There are several challenges that must be tackled to ensure contactless delivery of healthcare supplies through drones. First, the regulations associated with the usage of this technology are not crystal clear. There are social impacts; e.g., people feel afraid that the extensive applications of this technology can damage the labor market, as the transportation of the products does not need any manpower. Technical issues including the number of products, vendors, payload, and weather conditions have a great impact on the drone delivery industry.

**Figure 8 micromachines-13-01593-f008:**
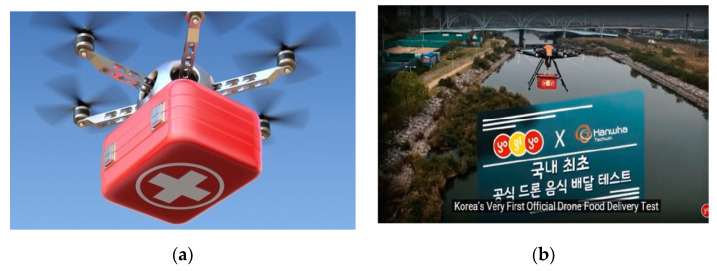
Drones used to deliver (**a**) medical supplies [63] and (**b**) food supplies [74].

**Table 7 micromachines-13-01593-t007:** Drone technology used for delivery services during the COVID-19 pandemic.

Reference	Country	Purpose
[75]	UK	To deliver COVID-19 tests, medicines, and personal protective equipment (PPE) to remote communities in the UK
[76]	Ghana	Zipline and UPS are working together to deliver COVID-19 vaccines to health centers in remote areas
[77]	USA	A North Carolina health system used UPS Flight Forward drones to deliver COVID-19 vaccines
[78]	UK	An NHS drone-based delivery service is used to carry personal protective equipment, blood tests, and COVID-19 samples
[79]	Indonesia	A group of drones has been used for contactless delivery of food and medicine
[80]	Scotland	UK-based drone specialist Skyports is providing drone service to deliver medical supplies and COVID-19 samples
[81]	China	Shanghai firefighters have been using drones to deliver medicines during lockdown
[82]	USA	Walmart has started a drone delivery service to collect kits in North Las Vegas and New York

### 4.6. COVID-19 Screening and Detection

A person-to-person encounter is one of the most frequent methods of coronavirus infection. Therefore, it is critical to quickly identify both the infected and potentially contaminated parties as a result of the interaction. Massive and quick screening in the surrounding areas is the plan, with little to no human-to-human contact. In this regard, drone technology has changed the game and aided in the mass screening of individuals based on symptoms [83]. These drones have a temperature sensor mounted to keep visitors away from their houses. Additionally, they have a camera for taking images of the subject and a GPS [84] for locating them. An overview of drone technology to find infected and healthy people in the streets can be seen in Figure 9.

In [1], the authors proposed Covidrone, which uses biomedical sensors to collect medical information from people linked to COVID-19. A sensor module, a disinfection module, testing kits, a wireless communication system, and a DNN model are included in Covidrone’s payload, as shown in Figure 10. Such drones could land in strategically chosen areas and wait for customers who needed their services again. Smart medical sensors on the drone acquire the necessary medical data after authenticating the user and disinfecting the sensing apparatus. Subsequently, a DNN-based model receives these medical data for COVID-19 identification and analysis. Based on the results of his or her test, the user will be advised to take additional medicine or take preventative measures. These test results would be accessible to remote medical workers via cloud services for additional analysis. The elimination of human touch due to process automation aids in the fight against COVID-19. This strategy can give older and physically challenged people access to crucial medical care and testing facilities. Covidrone is expected to help significantly in lowering COVID-19 curves, especially in remote areas where there are not many places to receive medical help. Table 8 summarizes the drone technology for the detection of COVID-19 in different countries.

To handle the COVID-19 outbreak, smart healthcare will be developed with the use of detection technologies. Drones, the Internet of Things (IoT), fifth-generation cellular networks, and a DNN model are some of these technologies. Drones have historically been employed in defense-related tasks, such as remotely operating aerial missile launchers. However, in recent years, drones have been utilized for a variety of purposes, including monitoring climate change, taking pictures and videos, and delivering blood. Researchers [85] suggest employing numerous UAV swarms to address various COVID-19 issues, including social isolation, symptom surveillance, and sanitization. For performance and security, they advocate the use of 6G, blockchain, and software-defined networking (SDN). Robots and drones were used by the authors of [9] to monitor COVID-19 hotspots and offer medical assistance where it would be required. Their platform also supports a number of COVID-19 activities, including sanitization, thermal imaging, crowd monitoring, and control. The viability of such systems’ operations is still in doubt because they may need extensive network resources and coordination when implemented in relatively broad locations, such as a village. In a recent study [86], the authors focused on drone-swarm technology to limit the spread of COVID-19. They proposed a system architecture that can identify the infected cases by estimating their biomedical parameters. They considered several techniques such as remote photoplethysmography, thermal measures, and stereoscopic vision to evaluate the target risk. The system can measure the image photoplethysmography signal, skin temperature, and social distancing, which are crucial parameters for tracing infected individuals.

**Table 8 micromachines-13-01593-t008:** Drone technology used for detection during the COVID-19 pandemic.

Reference	Country	Purpose
[87]	Australia	The Australian Department of Defense and the University of South Australia are working on drone technology to detect breathing, heartbeat, temperature, and infectious respiratory conditions through sensors and integrated cameras
[88]	India	Researchers from the Indian Institute of Technology (IIT) designed an IR-based drone for thermal screening and identifying COVID-19 suspected cases
[89]	China	Jiangxi province, China, used drones to check the temperature of people standing on balconies during lockdown
[90]	USA	In New York, Dragonfly’s pandemic drone is used to detect COVID-19 infectious conditions including coughing, sneezing, and temperature through integrated sensor and computer vision technologies

## 5. Emerging Technologies to Fight against COVID-19

This section discusses several emerging technologies which assist in fighting against COVID-19. At present, several technologies are being used to combat COVID-19, including blockchain [55,91], smart-phone applications [92,93,94,95,96,97,98,99,100], wearable sensing [101,102,103,104,105,106,107,108], artificial intelligence (AI) [109], robots [110], deep learning (DL) [111,112,113,114], virtual reality (VR) [115,116], and edge computing [117,118]. Figure 11 presents an overview of these technologies.

### 5.1. Blockchain

Blockchain technology is making substantial gains due to its proof of concept and use cases in different industries. Trust and robustness are the key features of this technology offering decentralized storage. It empowers both patients and medical service providers in a healthcare environment through secure data transactions. It can substantially remove physical labor for medical staff and considerably eliminates the infection risk. The integration of blockchain technology can significantly enhance epidemic prevention and control. This combination can ensure the reliability of collected data [91]. As most of the data are gathered from the media, public, and hospitals and are not privacy-preserved, blockchain technology causes COVID-19 data to be further improved and secured. It is not possible to further modify the transaction data stored in a blockchain. A blockchain can also secure data from cheating or hacking. It also ensures authentication and accountability to support data availability, integrity, and discretion. Moreover, a blockchain can properly work even if it partially fails. It offers several advantages such as transparency, immutability, and security. These features of decentralization and traceability make a blockchain unique as compared to other existing technologies. Thus, blockchain features can assist in data collection and perform further reliable analysis of COVID-19 data. Figure 12 outlines the use of blockchain technologies to tackle pandemic challenges through multi-drone collaboration [55]. The proposed architecture is based on four components. Different drones are used at different locations to perform various tasks. The blockchain can assist in securing each operation by keeping track of each drone’s operation. Each drone can access the locations of other drones, which can avoid collisions. In addition to the numerous advantages of utilizing blockchain technology to address COVID-19 pandemic concerns through multi-drone collaboration, there are a few limitations. The first are legal issues that must be resolved by various parties, such as international health organizations, country leaders, and international policymakers, in order to introduce new regulations regarding health policy; data sharing; digital health-service-related policy; and issues related to digital inequality, digital connectivity, and the digital divide that exists mainly in developing nations. Scalability is the second and most significant concern. Typically, drones have limited computational and memory capabilities [119]. As the number of transactions on the blockchain rises, the network’s traffic becomes increasingly heavy. Every node on the blockchain is required to keep all validated transactions, which presents a challenge given the block size and time interval required to generate a new block [120].

### 5.2. Wearable Sensing

Wearable technologies are developed with the integration of wearable sensors and the human body. These technologies can be found in the form of glasses, helmets, wristbands, and watches. These wearable technologies can perform various tasks in lifestyle, fitness, and healthcare [101]. In healthcare, they are used to monitor patients’ health records. IoT-enabled wearable devices include IoT-Q-Band [102], smart glasses [103], and smart helmets [104]. Figure 13 shows different wearable devices used for COVID-19. Recently, some studies have discussed the emerging wearable interface for human–drone integration to perform various tasks such as search and rescue [121], personal identification [122], and combatting pandemics [9]. The work presented in [122] can be used to trace people for monitoring, surveillance, and security purposes through drone technology. For instance, users, patients, or suspected cases can be integrated with wearable sensors such as smart bracelets connected to the internet. The IoT data can be retrieved from user wearables captured by a drone for personal identification.

In [9], the authors proposed a UAV-assisted smart healthcare framework for social distancing, sanitization, monitoring, and data analysis for COVID-19. The proposed system collects data through thermal image processing, mobility sensors, or wearable sensors. The gathered data are further processed through a multilayered framework for data analysis and decision-making. The proposed approach is demonstrated through simulation and implementation. In the implementation, it is stated that a large distance can be covered within a short time through this drone-assisted smart healthcare architecture for COVID-19. An overview of this proposed drone-assisted person monitoring architecture through wearable sensing technology is presented in Figure 14. Wearable sensors are used to monitor a patient under observation. Drones can collect patient data from these wearables, and the data are then stored in drone memory. Multiple servers are used to forward these stored data to big data. These servers use cloud, fog, and edge computing to process, model, profile, and analyze these data. After proper analysis, refined data are shared with hospitals through proper policies and regulations by the government and medical board. The data are securely transferred to medial staff whenever required. The proposed framework can be used to handle the COVID-19 pandemic situation. Regarding our discussion, wearable devices have demonstrated their promise during the COVID-19 pandemic; nonetheless, several limitations must be overcome. The majority of wearable devices are currently in prototype development. Integrating wearable devices into drones to aid in the COVID-19 epidemic is a significant problem in and of itself. In addition, issues such as consumer acceptability, security, ethics, and big data problems related to wearable technology must be addressed to improve the actual use and functionality of these devices.

### 5.3. Artificial Intelligence

Artificial intelligence (AI) is a powerful technique and tool which makes computers think and learn. This technology makes predictions of patterns. AI [109] has been extensively growing, and the research community has started integrating AI algorithms into drones and robotics for medical imaging [123,124,125] and other healthcare applications. AI can be implemented to predict the outbreak or spread of COVID-19. AI-based analysis models can be used to verify the statistical data about COVID-19. This can remove the healthcare staff burden of performing various duties such as the medical examination of patients [126]. Furthermore, AI can empower drones to enhance performance and productivity to fight against COVID-19 in multiple scenarios. COVID-19 has drastically enhanced the usage of AI in various environments. The potential of this technology can be validated in Scopus database where more than 1000 articles were published as of 1 March 2021 containing keywords “Artificial Intelligence and COVID-19” [5]. Like edge computing, AI has the potential to handle huge data. It makes use of big data analytics to deal with data. It can be integrated with UAVs to gather data. AI-empowered drones have been used to trace hotspot locations. UAVs and integrated sensors can gather data to perform necessary actions without human need. This promising technology has shown remarkable results in the diagnosis and treatment of infected patients as it can perform several tasks to offer substantial support to medical staff. Drones integrated with AI technology can support various features such as transportation, physical inspection, and surveillance [127]. In [128], the authors proposed an IoT/AI-based framework to monitor, track, and fight against COVID-19. The proposed IoT/AI-based system includes two components for monitoring and drone-assisted sterilizing. It includes a thermal camera that can be integrated into a helmet. The thermal or detected images are processed using an AI algorithm. The drone can be called once an infected case is detected. In a recent study [129], the authors proposed a conceptual model (shown in Figure 15). It is based on a drone, an AI framework, RF links, and a GCU. The proposed system can serve multiple tasks such as face mask detection, anti-social-distancing behavior, and violating rules for quarantine at home. In addition, it can also trace the activity of infected people and perform thermal imaging to check temperatures in order to stop the spread of the virus. Whether AI will be an effective tool against future diseases and pandemics depends heavily on data. The danger is that public health issues would override concerns about data privacy. It is possible that governments may continue their exceptional surveillance of civilians even after the epidemic has ended. Therefore, worries over the erosion of data privacy are warranted.

### 5.4. Deep Learning Algorithm for UAVs

In [111], researchers have proposed a DL-based drone technology to monitor the public and determine if someone is wearing a mask or maintaining social distancing. If anyone is not following the preventive measures, then it immediately sends a warning to the closest police station. The proposed approach utilizes a drone, integrated camera, sensors, and the Yolov3 algorithm to monitor public places. The proposed system also provides instructions to wear a mask. This system can save human lives by mitigating the spread of COVID-19 by monitoring social distancing. However, this system will cause implications in crowded places by capturing each individual’s image. Figure 16 illustrates the use of DL-based UAV technology to monitor public places. In a recent study [112], authors briefly explained the DL approaches for COVID-19 medical image processing. The authors also discussed the implementation of DL to control the outbreak and crises, ensuring smart healthy cities. In [113], the authors proposed a low-cost security mechanism for UAV-aided mask detection through DL and IoT in order to minimize COVID-19.

In a recent study [22], the authors proposed a deep transfer learning (DTL) real-time framework to identify overcrowding due to activities such as abnormal behavior and congestion. It was the first study that proposed a monitoring approach for the detection of overcrowding through a social monitoring system (SMS) and a UAV. The authors used DTL for decision-making and the water cycle algorithm (WCA) to significantly enhance the identification by choosing the best features. The proposed framework can operate in challenging environments to detect overcrowding through SMS communication and UAV video frames. An overview of this system is presented in Figure 17, which shows how overcrowding can be significantly controlled through UAV sensing and computing parts.

### 5.5. VR Technology

Virtual reality (VR) [130] refers to an extensive range of technical interventions. Those vary between large simulators, flat-screen projectors, and head-mounted displays (HMDs) [115]. They bring the users through computer-based graphics into an immersive multi-sensory scenario. Existing VR headsets offer affordable and safe rehabilitation at home. VR is used in a computer-based program to generate a preferable simulated environment. People can participate in any task in real time using a distributed whiteboard. This technology has become an integral part of telemedicine [131] as it plays a key role in enhancing medical knowledge, enhancing diagnostic skills, and fostering surgical skills. This technology has the potential to handle COVID-19 as it can provide awareness among people through image acquisition in order to alleviate the impact of the virus. VR provides a suitable environment for establishing video calls between doctors and patients in quarantine or hospitals. It can enhance the psychological side of patients, especially the quarantine patients, to give them confidence that the doctors are noticing all the vital signs or any possible complications of COVID-19 [132]. It can also ensure virtual visits which can reduce the patient’s need for assistance or the need for any family member to physically visit the hospital and reduces the cost of travel to the hospital. The key benefits of using this technology for COVID-19 are illustrated in Figure 18. This transformative technology can be integrated with UAVs to ensure a direct and broad view of activities. It is based on a headset integrated with sensors as shown in Figure 19. The operator can simply use head movements to control the UAV’s camera. The camera can scan the people, and video will be transmitted through a VR headset. This system can be used for real-time detection of people. Recently, a Canadian company, Dragonfly, has started working on drone technology as a screening tool to detect temperature and infectious respiratory conditions [133,134]. The healthcare staff can use this technology to remotely connect to, communicate with, and treat patients.

### 5.6. Edge-Computing-Based Drone Technology

IoT objects offer several services (autonomous driving, traffic management, location tracking, augmented reality (AR), etc.) which usually need extensive data-processing, leading to high computational resource demands. Mobile edge computing (MEC) is proposed to effectively overcome this challenge. MEC servers can offload extensive computational tasks from IoT objects. They integrate edge devices for real-time data processing through sensors and microcontrollers or microprocessors. However, delays and communication overhead are still critical concerns. With the potential of low cost and high mobility, UAVs can overcome these challenges by operating as MEC servers [117]. The offloading decisions for such systems include execution delays, power consumption, and service latency. In a recent work, authors have addressed the incorporation of UAVs and edge computing which can mitigate the challenges of onsite computation, data processing, connectivity, and latency [118]. The proposed approach can be used in containment zone detection and thermal sensing. The proposed idea creates the possibility of using drones to explore the COVID-19 pandemic smoothness for some social duties. In this system, edge computing plays a major role in strengthening data processing without utilizing bandwidth over the internet. It considers the flexible deployment of edge nodes and multiple drones using 5G spectra. The proposed system architecture in shown in Figure 20. Multiple drones can be linked for data storage or communication with a central cloud server. This server is connected with the users in that specific area where detection is being performed. This approach can be used on streets to deploy numerous drones for COVID-19 victim detection. Table 9 summarizes the use cases of drones for combating pandemics with the integration of different emerging technologies.

## 6. Open Challenges, Opportunities, and Future Work

There are several challenges and security issues that must be addressed in future works related to UAV-based COVID-19 applications.

Patient Health History: The existing works focus on camera-based UAVs for scanning and capturing the user for authentication tasks. In the future, researchers can use advanced processing tools and cameras which keep a record of previous history and reports of the patient. This can significantly assist medical staff in knowing previous conditions and recommending medical advice accordingly.Reconfigurable Platforms: There is a need to integrate reconfigurable platforms such as FPGAs which can add, modify, or remove any function virtually. This can ensure availability and flexibility and enhance performance. These platforms can assist in optimal resource allocation and remove the burden of some hardware payloads.Integration with Healthcare Databases: Future studies should focus on integrating UAV-assisted COVID-19 screening and detection systems with national or international health monitoring databases. This can ensure real-time and accurate updates about positive cases, asymptomatic cases, active cases, and number of deaths; highlight high-risk areas; and assist regulatory bodies in taking immediate actions to contain the spread of disease through proper planning such as implementing lockdowns.Drone Navigation in Challenging Environments: In some scenarios, the UAVs are required to operate in densely populated areas or thick forest-covered areas. Uneven atmospheric conditions such as haze, fog, and rain can also severely impact the performance of UAVs. To overcome these issues, future works can incorporate computer-vision-based strategies for the safe operation of UAVs. There should be proper insurance liability due to damages caused by UAVs. Several media reports describe soft tissue injuries, eye loss, and severe lacerations due to UAV accidents. In addition to property damage and injuries due to UAV crashes, UAVs also cause accidents with aircraft and liability due to damaged goods and dropped cargo. Liability for UAV use also contains an enormous threat to individual privacy.Health and Safety Risks: Using drones in medical facilities where they need to operate in close proximity to patients generates safety and health concerns. For instance, drone batteries must be safe and sealed to prevent any chemical damage. The drone rotors or wings must be guarded to avoid any accidents which can cause severe injuries. Future works should address these issues for the safe operation of UAVs.Security, Privacy, and Safety Concerns Associated with Drone Use: Drones deployed in densely populated places, especially during COVID-19 and other pandemics, may create security, privacy, and safety concerns. Drones crashing on public land might pose a safety risk. This might happen due to a number of factors, including human error, a breakdown in machinery, a collision in flight, or a lack of proper maintenance. In addition, there is a significant danger of airborne accidents resulting in extensive devastation because of the congestion of airspace in bigger cities from commercial flights. Drones equipped with high-resolution cameras, sensors, and recorders can be piloted remotely for pinpoint surveillance. Using a drone equipped with malicious software, it is possible to collect sensitive information, monitor people’s movements, and create detailed profiles about them using wireless localization. Moreover, the navigation and communication units of drones are susceptible to a variety of security vulnerabilities [152]. Due to the open nature of unencrypted and unauthenticated GPS signals, they are easily spoofable [153]. GPS allows the navigation system of a drone to function [84]. Wi-Fi jamming is another conceivable assault that might result in the loss of control of the drone’s communication system, which would have severe repercussions for nearby individuals. Concerning is the fact that such attacks may be carried out using commercially available products. If the drone is running on an unprotected Wi-Fi network, a normal cell phone may suffice. In addition, as the number of drones continues to increase and wireless technologies continue to advance, security and privacy issues are growing. In addition, AI/ML and Edge-AI/ML-based UAV capabilities on drones need the implementation of security measures to prevent AI-related attacks. Model inversion and model extraction are two ML-as-a-service threats recently uncovered in drones. The training data are recreated from the model parameters via model inversion attacks to retrieve sensitive and private information. Model extraction attacks are dependent on model parameters obtained via model querying. Drones connected to 6G networks and supporting technologies can help mitigate the aforementioned challenges. Additionally, lightweight authentication methods, federated learning, and aerial blockchain technology will enhance the cyber-physical attack defense of drones.Standardization: Despite the extensive emergence of UAVs, there is a dire need to devise standardizations from regulatory bodies for the operations of UAVs in geographic areas of different countries. A major hindrance in the widespread use of UAVs is the ambiguity or lack of significant standards and regulations for UAV operations, allowed airspace, allowed weight and size, allowed height, privacy or secrecy considerations, safety requirements, and characteristics. A lack of homogeneity in government rules for the implementation of UAVs can be observed. UAVs can affect the navigation of commercial airplanes. So, countries should implement regulations and rules for the proper operations of UAVs. For instance, in the US, the FAA is responsible for issuing certificates and air traffic regulations for UAVs. Similarly, international collaboration or coordination can also assist in introducing global rules, as different countries have different regulations and standards.Resource Allocation: Due to UAV path constraints and battery limitations, resource allocation has become a crucial concern. It is noticed in three aspects: UAV hovering, local computing, and task offloading. Thus, designing accurate path planning can consequently compromise the operational cost and calculated performance. Intelligent deployment [154] and efficient resource allocation can improve fairness, coverage, task completion time reduction, cost reduction, power consumption reduction, and computational efficiency maximization.Limited Energy: Another hurdle for UAV operation performance is power limitation, energy consumption, or limited battery life. Usually, UAVs are battery-powered and suffer from short battery life, generally below 1 h. UAV batteries are consumed for image analysis, data processing, wireless communication, and UAV hovering. Usually, UAVs need to travel over large areas and need to return multiple times to charging stations. In SAR operations, UAVs fly for longer time periods over disaster-stricken areas. Due to these limitations, a decision should be made as to whether UAVs carry out image or data analysis in real time or not. One possible solution is to form swarms of drones through coordination algorithms that can overcome the limitation of a single drone in terms of energy efficiency. Throughout the COVID-19 outbreak, drone swarms were employed for a wide range of purposes. While the majority of such drones were operated by humans, studies on drone swarm control schemes have demonstrated that autonomously piloted drones outperformed personnel in a number of activities. A recent study lays the groundwork for a paradigm that uses variable independence in drone handling during outbreak missions to improve overall performance. It also discusses the issues to consider while creating mixed-initiative platforms that swing between human control and autonomy during a task. Flexible autonomy guarantees that individuals who comprehend the pandemic’s social characteristics are involved in the control loop [155]. Other interesting approaches are the investigation of novel designs for recharging stations and the use of efficient wireless power transfer (WPT) methods such as laser power transfer (LPT).Connectivity: It is critical for a FANET to provide network connectivity to drones and to or from other devices in order to be considered as a communication network. To stay connected, deployed UAVs must fly inside the covered region. However, the drone’s low battery is an impediment to a long-term mission, which might result in traffic gridlock with no access possibility. As a result, determining whether to add an innovative UAV to the UAV swarm or to remove a drone (due to low battery) is an issue to be addressed.Atmospheric Conditions: In case of adverse weather conditions such as storms, rain, and wind, UAV deployment for different applications such as precision agriculture is difficult due to unwanted deviations in predetermined trajectories. Weather conditions also affect operation time, path elevation, UAV altitude, and flight direction. In natural disaster conditions, e.g., typhoons, hurricanes, or tsunamis, atmospheric conditions tend to be a cardinal challenge for UAV missions. In these detrimental conditions, UAVs cannot hover and cannot collect accurate readings or data, and they cannot operate in extreme conditions. Therefore, researchers should address the specifications and UAV capabilities to withstand these adverse weather conditions and complete weather-sensitive missions efficiently and safely. Specifically, wind speed should be taken into account for smooth UAV operations, and it should be involved in the UAV strategic mission plan and deployment phase.Privacy: Another major concern about privacy arises with the use of UAVs. UAVs are incorporated with cameras or other equipment that can capture photos or record videos; which may result in the violation of individual privacy. To tackle this problem in the USA, the Center of Democracy and Technology (CDT) instructed Federal Aviation Administration (FAA) to develop specific regulations to preserve privacy. For this purpose, Privacy by Design (PbD), which supports compensations for privacy violations, was introduced [156]. PbD regulations notably restrict privacy intrusion. Furthermore, some countries, e.g., China, do not have specialized privacy protection regulations for drones. This requires drones to be updated as quickly as feasible in order to design appropriate requirements for drones. Privacy is a valuable personal right, yet it is rarely given much consideration in China. In reality, drones are not prevalent in infringing on private rights. Drones having photographic features, for example, jeopardize the privacy of people who are caught unaware by such cameras [157]. Moreover, contact tracking tools can assist in containing the spread of the virus, but these tools also invade people’s privacy. Smartphone-based COVID-19 tracking applications can access individuals’ information and location. The wristband for tracking COVID-19 patients can also cause privacy issues. Access to such sensitive information should not be given to any unauthorized user. There must be proper agreement or guidelines for data collecting and sharing and keeping secrecy. Furthermore, monitoring people for social distancing also raises privacy concerns. Aerial surveillance can damage an individual’s freedom and privacy.Cyber Security: COVID-19 has raised cyber security threats besides jeopardizing the economy and healthcare sectors. This is occurring due to the reliance on people to work remotely during lockdowns, which makes them vulnerable to malicious threats, such as fake vaccine products, fake health insurance, malware in COVID-19 resources, and phishing emails. Cyber security concerns can also impact industrial automation, e-commerce, and online business sectors. Thus, it is essential to develop innovative cyber security tools to ensure robust, reliable, normal, and safe operations.Telehealth: CVOID-19 has proven to be a game-changer for several economic and industrial sectors such as healthcare [158] and the mask industry. There is an expanding upsurge in healthcare systems, operations, and users. It is difficult to handle this sudden rise without reliable and efficient wireless connectivity. AI, AR/VR, robotics, and IoT technologies are envisioned to revolutionize the healthcare industry. These technologies will propose innovative strategies such as remote surgery. These technologies can also substantially reduce travel costs as patients can easily access healthcare professionals online. Thus, telehealth can support patients who are unable to visit hospitals in person.A More Connected World: The COVID-19 pandemic is envisaged to revolutionize inter-connected technologies across the globe. There is a notable rise in the use of several applications for online conferencing, audio/video calls, and online classes. These applications, including Skype, Zoom, DingTalk, Tencent Meeting, and several other tools, are being used for e-learning and teleconferencing. People have become highly dependent on these tools, which can lead to a remote workforce in the future. Furthermore, the integration of other promising technologies such as 5G/6G, IoT, and ML/DL will pave a way for virtual conferencing, classrooms, hospitals, and labs.The IoT Revolution: During COVID-19, there has been a notable increase in IoT networks, consequently leading to the automation of various sectors such as transportation, consumer electronics, manufacturing, and health. With the integration of 5G and future 6G technologies, IoT is expected to transform the current world into a smart world of smart cities, smart oceans, smart hospitals, smart factories, smart spaces, etc. The ongoing research activities in this domain can help the research and medical communities to adopt innovative strategies to handle any pandemic or calamity in the future.While making decisions about counter-drone technologies, security professionals, law enforcement agencies, technology developers, and government officials mostly consider two factors: cost and efficiency. The vast societal impacts are mostly disregarded. Ignoring economic aspects can lead to market loss. Ignoring efficiency can lead to users’ bad response and less trust in adopting this technology, which can ultimately decrease market value. It can lead to psychological stress for both users and developers [159]. A recent study discusses the emotional evaluation of drone safety and perception of risk during the COVID-19 outbreak in Italy [160]. It is very essential to estimate safety and risk for security and safety management. For this purpose, the use of opinion mining and sentiment analysis to ensure public security and safety has attracted remarkable interest.Reliability: Because of erratic topological changes, UAV networks might encounter malicious nodes or a bottleneck relationship between two source destination sets. A standard approach for ensuring reliability is to provide an alternative channel between a particular destination and a given source via a group of UAV relays. Nevertheless, the great mobility and short battery lifespan of UAVs make such a system ineffective [37].Artificial intelligence faces an obstacle concerning human–machine interaction. This domain is at the crossroads of several AI disciplines and must identify them holistically: modeling humans and human cognition; procuring, portraying, and exploiting abstract knowledge at the personal level in an amenable fashion; explanation of this information needed to make decisions; and ultimately instantiating those choices into physical actions both legible to and in cooperation with humans. Human–computer interaction (HCI) experts endeavor to transform breakthrough technologies such as quantum computing and AI into attractive, user-friendly human–machine interaction interfaces that are pertinent to our daily lives [161]. Natural language processing (NLP) is an essential approach for increasing the comprehension of human–machine interactions. The credibility of designed systems and their acceptance by human users are major concerns for AI professionals. Additionally, they exhibit certain patterns that demonstrate how this structure supports the administration and operation of associated learning pathways and model training while bringing verified AI laws. The surveillance and information accumulation unit, the trust computation and assessment unit, the trust recommendation unit, the decision-making unit, and the diffusion of the detection unit are the five basic parts of trust-based strategies. Currently, a trust-based approach creates a reputation system made up of many parts to offer security from route assaults. Within the trust monitoring/development process, it is possible to identify abnormalities and misbehavior of drones in the system. For a job stated in the swarm, offline/online collision assessment/detection and stochastic models that evaluate collisions can be employed. Unfortunately, the edge analytic capacities are limited by end-to-end latency requirements and the severe limits of the swarm drones’ systems. Verified scalable holistic abstraction can reduce latency in bottlenecks by automating the life cycle of AI interactions.

## 7. Conclusions

During the COVID-19 pandemic crisis, healthcare practitioners and researchers have investigated technological approaches for combating the virus. Researchers from a number of disciplines presented a variety of techniques and possibilities to combat this epidemic. In this light, drones have shown to be a useful instrument in the battle against COVID-19 and have demonstrated tremendous promise by delivering speedy services. Drone technology may become one of the best friends in the fight against COVID-19-like diseases in a future of enforced social isolation. This research investigates drone technology via the use of real-world examples and applicable application situations. Drones may be used with several modern technologies to allow dispersed processing features. This paper also discusses briefly a number of challenges and opportunities notwithstanding this fact. Several use cases of drones for reducing the risk of COVID-19 are discussed in this study, including surveillance, inspection, message broadcasting through QR codes and loudspeakers, medical supply delivery, disinfectant spraying, patient screening, and patient identification. In addition, the authors have discussed a number of emerging technologies, including artificial intelligence, machine learning, deep learning, blockchain, the Internet of Things, edge computing, and virtual reality, which offer a variety of technical innovations to manage this disease with minimal human intervention. The authors finished this research with the hope that it would help academics and professionals overcome the many obstacles that make it hard to use UAVs in these kinds of situations.

## Figures and Tables

**Figure 1 micromachines-13-01593-f001:**
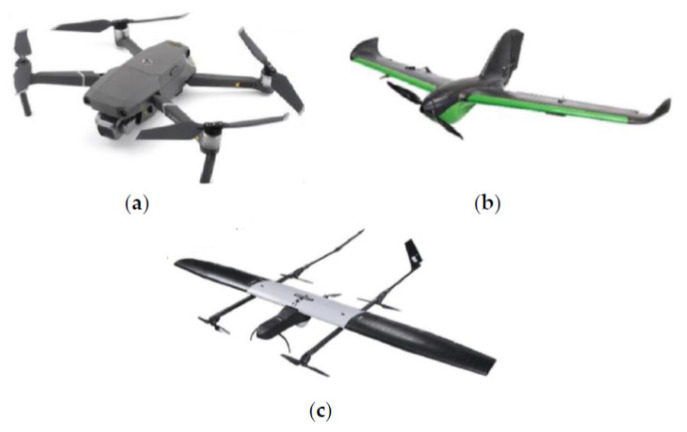
Different types of UAV: (**a**) rotary wing, (**b**) fixed wing, (**c**) fixed-wing hybrid [35].

**Figure 3 micromachines-13-01593-f003:**
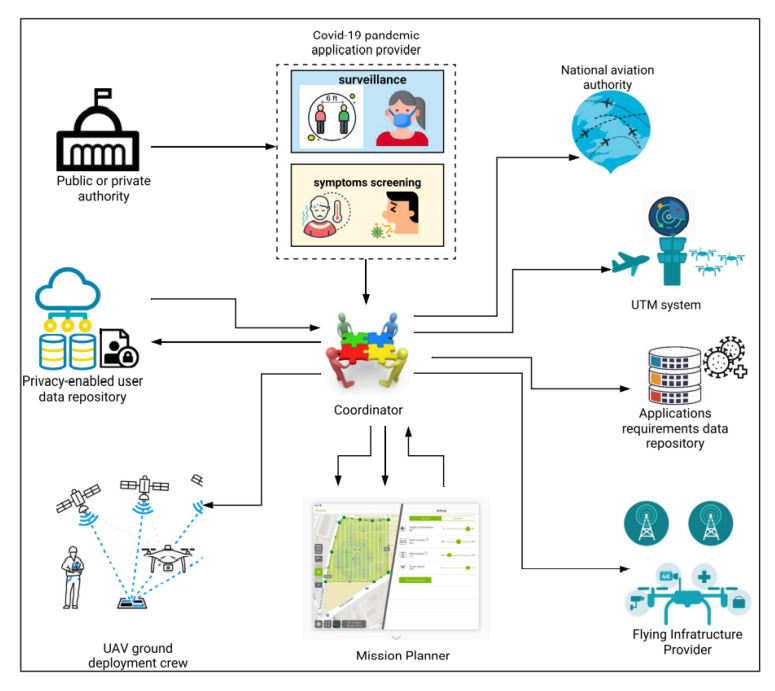
UAV swarm in COVID-19 pandemic [37].

**Figure 5 micromachines-13-01593-f005:**
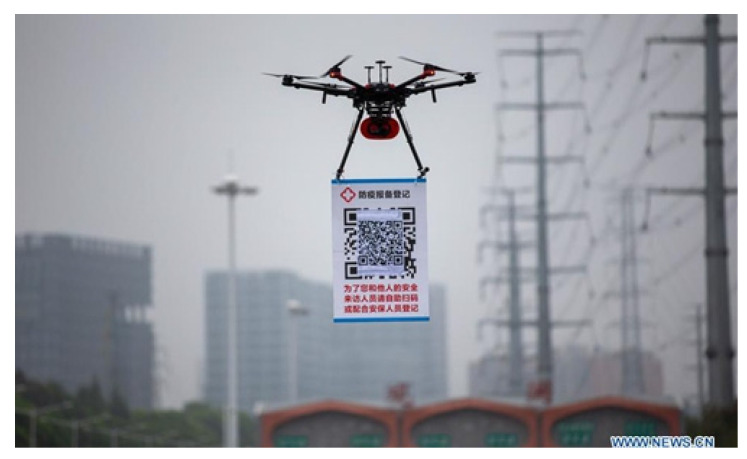
Broadcasting a message through a QR code [51].

**Figure 6 micromachines-13-01593-f006:**
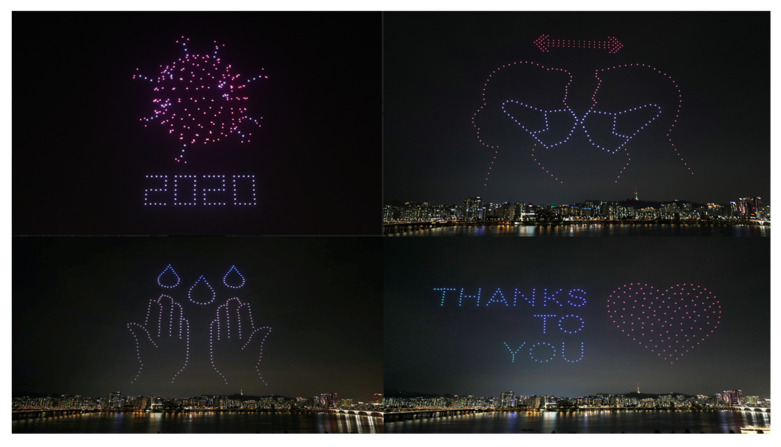
Broadcasting messages through the illumination of drone swarms in South Korea [53].

**Figure 9 micromachines-13-01593-f009:**
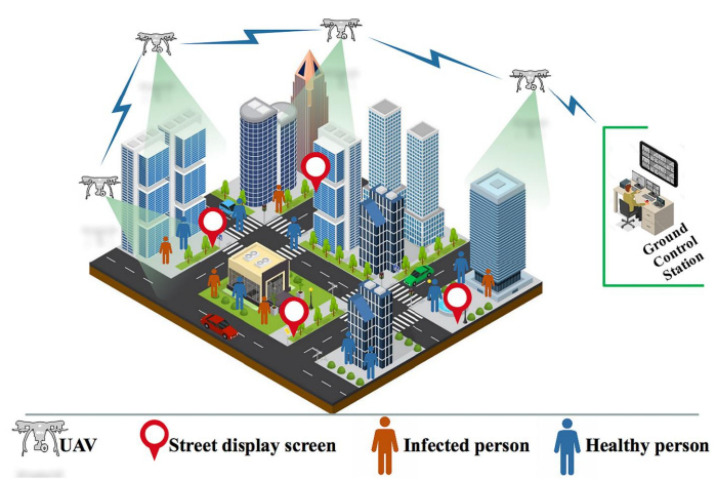
Drone technology used for the detection of infected people [37].

**Figure 10 micromachines-13-01593-f010:**
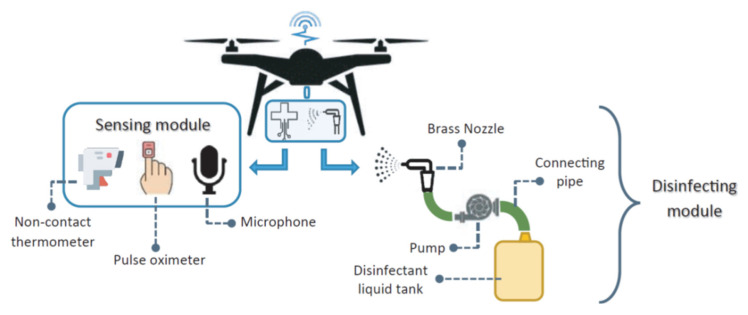
Covidrone schematic for sensing and disinfection [1].

**Figure 11 micromachines-13-01593-f011:**
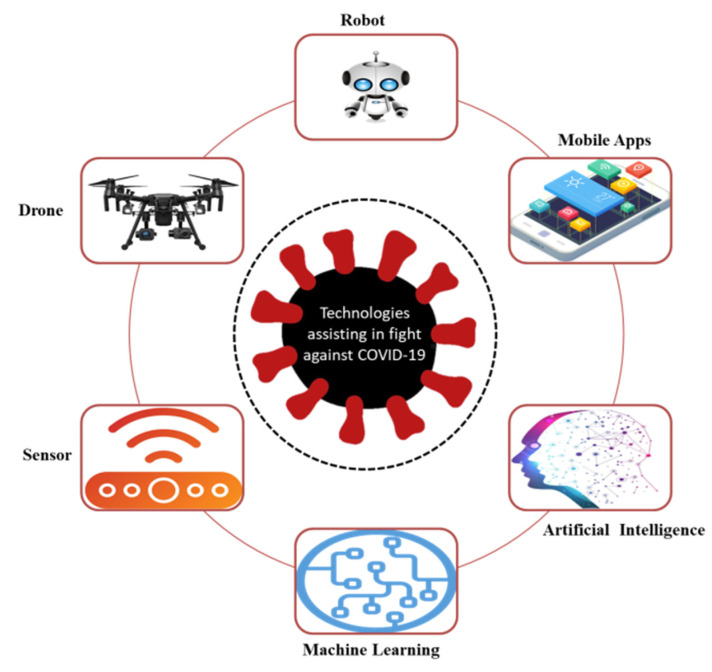
Some ingenious technologies to fight against COVID-19.

**Figure 12 micromachines-13-01593-f012:**
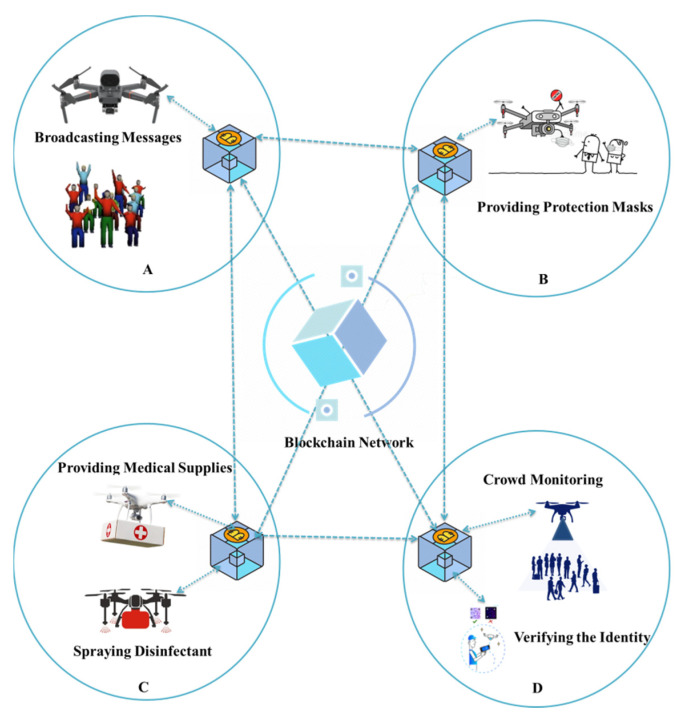
Application of blockchain for multi-drone collaboration: (**A**) broadcasting messages, (**B**) providing protection masks, (**C**) providing medical supplies and spraying disinfectant, (**D**) crowd monitoring and verifying the identity (modified from [55]).

**Figure 13 micromachines-13-01593-f013:**
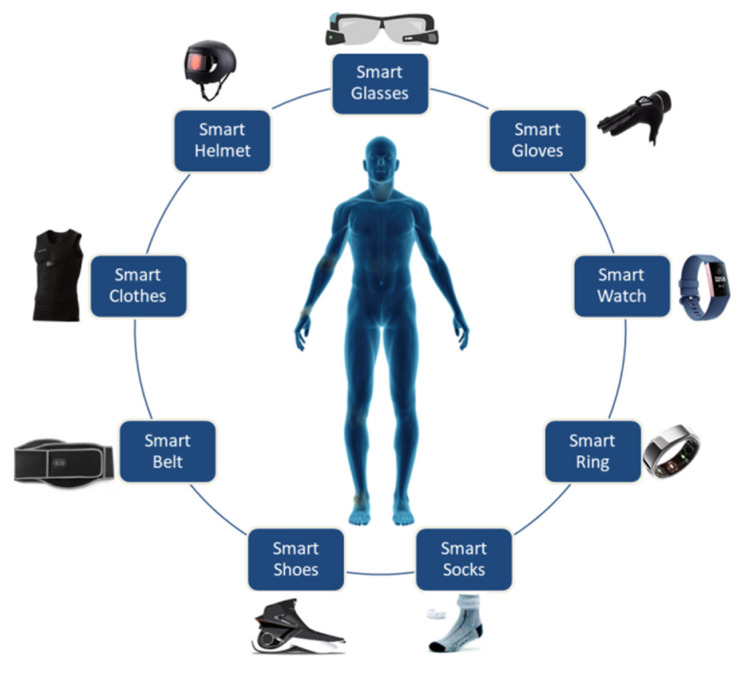
Various wearable sensing devices for COVID-19.

**Figure 14 micromachines-13-01593-f014:**
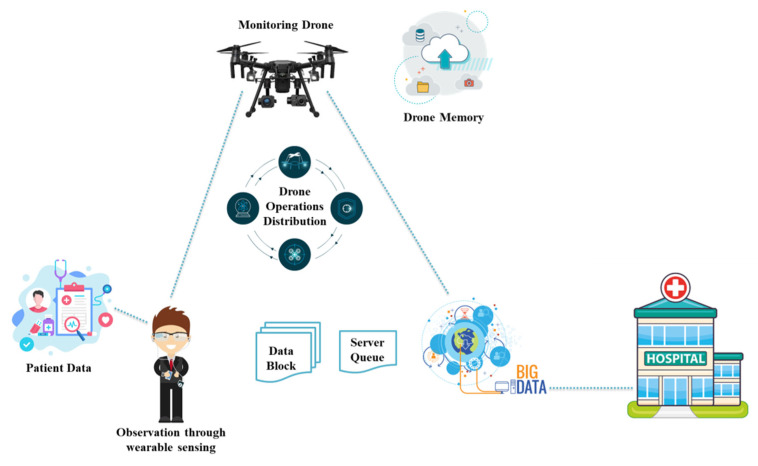
Drone-assisted person monitoring system using wearable-sensor data (modified from [9]).

**Figure 15 micromachines-13-01593-f015:**
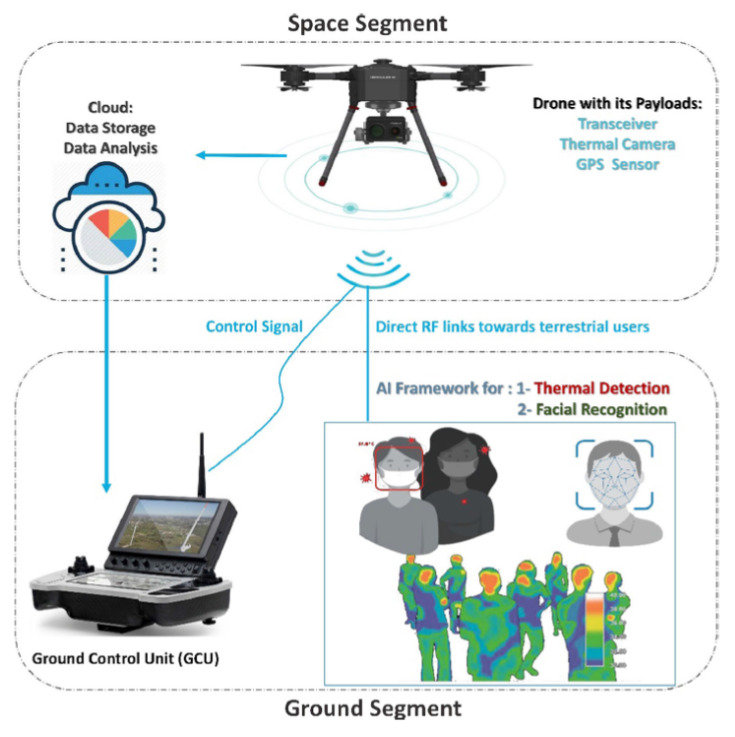
Conceptual architecture of the proposed model AI framework [129].

**Figure 16 micromachines-13-01593-f016:**
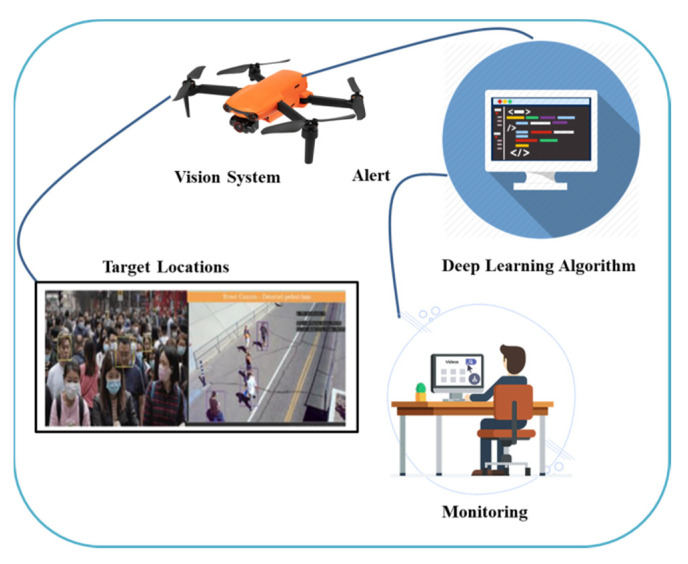
DL-based UAVs to monitor public places (modified from [114]).

**Figure 17 micromachines-13-01593-f017:**
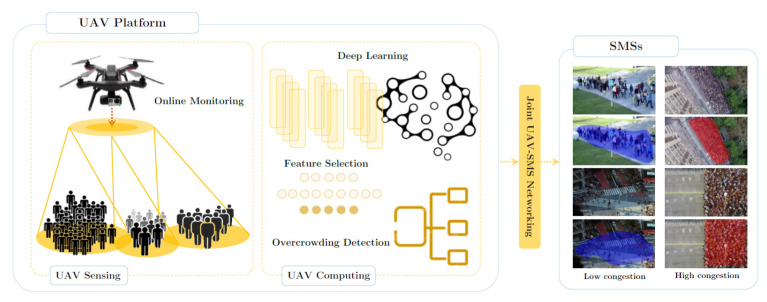
UAV and SMS to control COVID-19 spread [22].

**Figure 18 micromachines-13-01593-f018:**
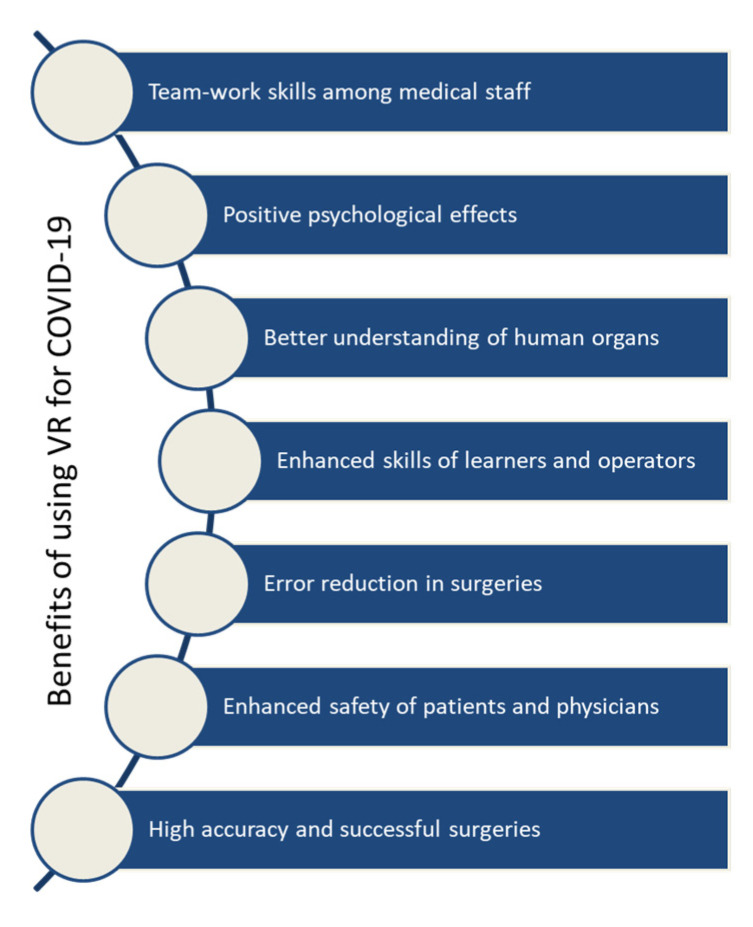
Key benefits of using VR technology for COVID-19 (modified from [135]).

**Figure 19 micromachines-13-01593-f019:**
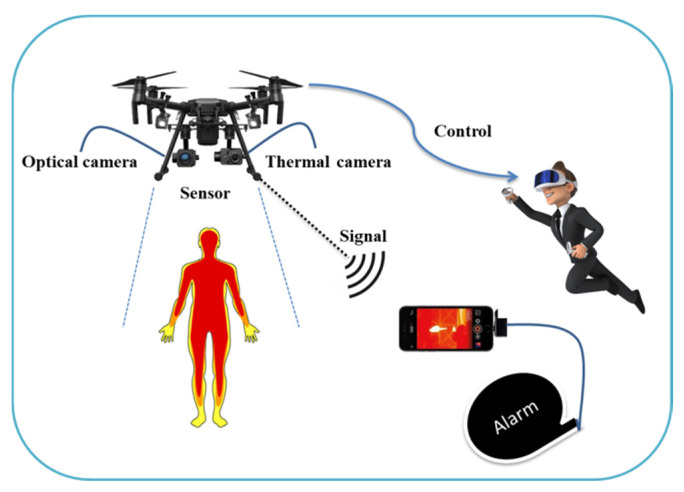
VR-based monitoring through UAV (modified from [116]).

**Figure 20 micromachines-13-01593-f020:**
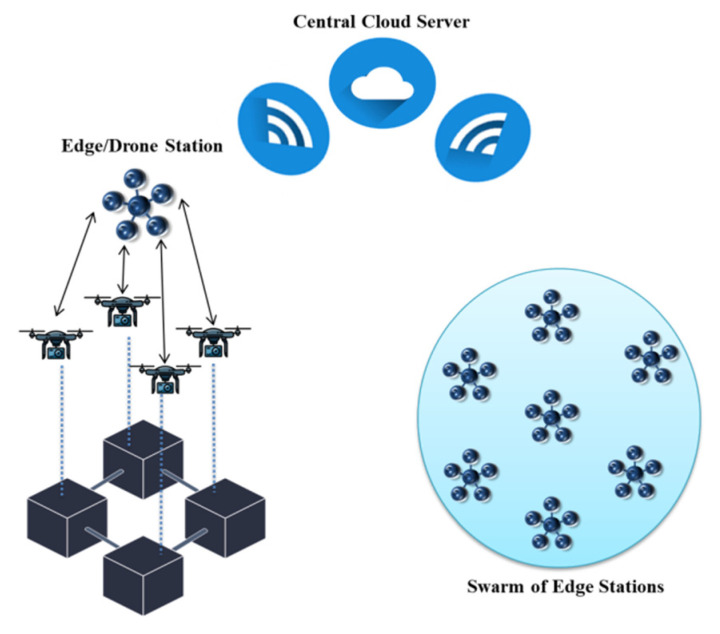
Edge–drone application scenario (modified from [118]).

**Table 1 micromachines-13-01593-t001:** UAV-based system for pandemic or disaster analysis.

Reference	Year	System Purpose
[15]	2018	UAV-aided system for medical applications
[16]	2018	UAV system for post-disaster operations
[17]	2019	UAV-based system for healthcare use cases
[18]	2019	UAV system to deliver medical drugs
[19]	2020	UAV system for medical supply during disaster in Japan
[9]	2021	UAV-based system to combat COVID-19
[20]	2021	UAV-based system to monitor social distancing
[21]	2021	UAV-based system to supply medical resources during epidemics
[22]	2022	UAV-based smart visual sensing for overcrowding
[23]	2022	UAV-based isolation control proposal for COVID-19

**Table 2 micromachines-13-01593-t002:** Applications, advantages, and disadvantages of different types of UAVs.

UAV Type	USD Price	Applications	Advantages	Drawbacks
Fixed wing	USD 20,000–150,000	Structural inspection, area survey	Large area coverage, long endurance, high speed	Launching, landing, high price
Rotary wing (helicopter)	USD 20,000–150,000	Supply drops, inspection	Hovering, large payload	High price
Rotary wing (multicopter)	USD 3,000–50,000	Photography, filmography, inspection	Hovering, availability, low price	Short flight time, small payload

**Table 3 micromachines-13-01593-t003:** Characteristics of different UAVs.

Characteristics	Fixed Wing	Rotary Wing	Hybrid
Energy efficiency	High	Low	High
Flight system	Complicated	Simple	Complicated
Landing	Conventional	Vertical	Vertical
Autonomy	No	Yes	Yes
Hovering	No	Yes	Yes
Power supply	Battery, fuel	Battery	Battery, fuel
Endurance	60–3000 m	6–180 m	180–480 m
Payload	1000 kg	50 kg	10 kg
Weight	0.1–400,000 kg	0.01–100 kg	1.5–65 kg

**Table 4 micromachines-13-01593-t004:** Drone technology used for surveillance during the COVID-19 pandemic.

Reference	Country	Operation
[38]	Kazakhstan	To patrol and monitor illegal border activities to stop the spread of COVID-19
[39]	France	To patrol over a closed beach to monitor public
[40]	Australia	To monitor social distancing at the beach
[41]	India	To ensure lockdown implementation by monitoring public
[42]	Spain	To monitor streets for anyone ignoring COVID-19 lockdown

**Table 9 micromachines-13-01593-t009:** Use of drones with different integrated technologies.

Reference	Integrated Technology	Use Case
[1]	Internet of Mobile Things, Deep Neural Network	Rapid testing, quick delivery of medical kits
[9]	Wireless Sensor Networks	Sanitization, thermal image collection
[20]	YOLO-v3	Social distancing
[21]	Machine Learning	Medical source distribution
[22]	Visual Sensing, DTL	To monitor overcrowding in order to control COVID-19 spreads
[118]	Edge Computing	Containment zone detection with offline data processing and computation
[131]	Could Virtual Reality	Role in the rehabilitation after COVID-19 infection
[136]	IoT	Spread reduction
[137]	Blockchain, 5G networks	Vaccine distribution
[138]	Blockchain, 6G networks	Mass surveillance
[139]	Internet of Things	Sanitization
[140]	Blockchain, AI	Pandemic supervision
[141]	Convolution Neural Network (CNN)	Crowd density
[142]	Deep Learning	Fast detection of medical face masks
[143]	YOLO-v3	Localization, navigation, people detection, crowd identification, and social distancing warning
[144]	Hybrid Reinforcement Learning	Optimized path planning for UAV COVID-19 test kit delivery system
[145]	IoT	Identifying COVID-19 patients for providing medical services using drones
[146]	6G and Blockchain	Proposes a generic scheme to integrate UAV, 6G, and BC for secured vaccine distribution
[147]	Machine Learning	Crowd surveillance
[148]	AI, IoMT	For monitoring and supplying COVID-19 patients
[149]	Radar Sensing	Detection of respiratory disorders in COVID-19 patients
[150]	Blockchain	Secure and QoS-aware drone delivery framework
[151]	Reinforcement Learning, Wireless Network	COVID-19 tracing

## Data Availability

Not applicable.

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
