# Peer review of "Role of Drone Technology Helping in Alleviating the COVID-19 Pandemic"

_micromachines, 2022, doi:10.3390/mi13101593_

Round 1
Reviewer 1 Report
This is an interesting work and worthwhile use case for drones. My main question is this:
Have the authors considered how to deal with potential misuse or attempts to sabotage these systems? How to deal with the security of the drones when they are operating in public?
Author Response
Dear Reviewer,
Thank you for your valuable time in giving comments to improve the quality of this article.
Please find the point-by-point author's response in the attachment.
Regards,
Authors

Reviewer 2 Report
The abstract does not clearly show the methodology. The main findings and the main conclusion. The summary goes a long way in the importance of the topic.
From section 2 onwards, the structure of the text is not properly understood, it seems like an extensive literature review but the common thread is not clear.
It is necessary to add a detailed and replicable methodological section. It is suggested to raise it from Systematic Review of Literature (PRISMA or other), Bibliometry, Meta-analysis or other structured.
Not being able to specify search, delimitation, inclusion/exclusion criteria does not allow for robustness or verification of the quality of the suggested publications, nor that significant contributions have not been excluded.
Interesting information is presented but it is not observed if it responds to a research question or problem, only an extensive text with many references is structured.
The objective of the article is to show the contribution of drones to help alleviate the pandemic and it is vital that the research problem is articulated with the questions that the review answers.
Very good contributions are observed regarding the various options for the use of drones and successful experiences reported in other articles. This reflects great input and high citation potential.
It is necessary to clarify two elements in the part of the research agenda: the contribution to knowledge of this review and the practical implications of the results found.
Author Response

(The authors gave the same response as above.)

Reviewer 3 Report
This paper discusses various unmanned aerial vehicles (UAVs) applications to support healthcare systems. The main contribution of the paper is a detailed review of practical approaches for combating COVID-19 through the integration of promising technologies including Artificial Intelligence, the Internet of Things, Blockchain, Virtual Reality, and Edge Computing.
Nevertheless, there are a number of remarks that should be brought to the attention of the authors
1. In Sub-sections 5.1-5.6, it would be advisable to show not only the possibilities of the considered technologies (Blockchain, Wearable Sensing, etc.) but also their limitations for fighting against COVID-19
2. It would be appropriate to supplement Table 9 with use cases for virtual reality and edge/fog/cloud computing-based drone technologies
3. In Section 6 (Attacks), the authors should consider not only attacks on individual UAVs but also attacks on the networks they form. It is also important to mention onboard intrusion detection systems (IDS), including those that use AI techniques.
4. As the authors consider UAV-based COVID-19 applications utilizing AI methods, trustworthy AI issues should be mentioned in Section 6.
5. In Section 6, it would be appropriate to consider UAV and UAV-based solution reliability issues.
6. The size of the explaining inscriptions in Figures 5 and 8 should be big enough to be legible.
7. To make the text of the paper more understandable in the context of the used abbreviations, the last can be presented as the List of Abbreviations in Appendix A.
Author Response

(The authors gave the same response as above.)

Round 2
Reviewer 2 Report
The authors have addressed all suggestions adequately